# Effect of Essential Oils on the Oxyntopeptic Cells and Somatostatin and Ghrelin Immunoreactive Cells in the European Sea Bass (*Dicentrarchus labrax*) Gastric Mucosa

**DOI:** 10.3390/ani11123401

**Published:** 2021-11-29

**Authors:** Maurizio Mazzoni, Giulia Lattanzio, Alessio Bonaldo, Claudio Tagliavia, Luca Parma, Serena Busti, Pier Paolo Gatta, Nadia Bernardi, Paolo Clavenzani

**Affiliations:** 1Department of Veterinary Medical Sciences, Alma Mater Studiorum—University of Bologna, Ozzano Emilia, 40064 Bologna, Italy; giulia.lattanzio@studio.unibo.it (G.L.); alessio.bonaldo@unibo.it (A.B.); claudio.tagliavia2@unibo.it (C.T.); luca.parma@unibo.it (L.P.); serena.busti2@unibo.it (S.B.); pierpaolo.gatta@unibo.it (P.P.G.); paolo.clavenzani@unibo.it (P.C.); 2Tecnessenze s.r.l., Minerbio, 40061 Bologna, Italy; n.bernardi@tecnessenze.com

**Keywords:** essential oil, oxyntopeptic cells, Na^+^K^+^-ATPase, somatostatin, ghrelin, enteroendocrine cells

## Abstract

**Simple Summary:**

Nutritional strategies focused on the use of botanicals as modulators of several physiological responses and health promoters of the gastrointestinal tract have attracted interest in animal production. Previous research indicates the positive results of using essential oils (EOs) as natural feed additives for several farmed animals. In the last decades, these nutritional alternatives have been evaluated and reported in fish production in order to increase fish growth and feed utilization and to promote animal welfare. Therefore, the present study was designed to compare the effects of feed EO supplementation in two different forms (natural and composed of active ingredients obtained by synthesis) on the gastric mucosa in European sea bass. EOs decrease oxyntopeptic cells and increase somatostatin and ghrelin enteroendocrine cells. In addition, we showed that Na^+^K^+^-ATPase was expressed in oxyntopeptic cells (OPs) in the same way as H^+^K^+^-ATPase (typical marker for mammalian parietal cells) and, for this reason, consider Na^+^K^+^-ATPase a valid marker for OPs.

**Abstract:**

The current work was designed to assess the effect of feed supplemented with essential oils (EOs) on the histological features in sea bass’s gastric mucosa. Fish were fed three diets: control diet (CTR), HERBAL MIX^®^ made with natural EOs (N-EOs), or HERBAL MIX^®^ made with artificial EOs obtained by synthesis (S-EOs) during a 117-day feeding trial. Thereafter, the oxyntopeptic cells (OPs) and the ghrelin (GHR) and somatostatin (SOM) enteroendocrine cells (EECs) in the gastric mucosa were evaluated. The Na^+^K^+^-ATPase antibody was used to label OPs, while, for the EECs, anti-SOM and anti-GHR antibody were used. The highest density of OP immunoreactive (IR) area was in the CTR group (0.66 mm^2^ ± 0.1). The OP-IR area was reduced in the N-EO diet group (0.22 mm^2^ ± 1; CTR vs. N-EOs, *p* < 0.005), while in the S-EO diet group (0.39 mm^2^ ± 1) a trend was observed. We observed an increase of the number of SOM-IR cells in the N-EO diet (15.6 ± 4.2) compared to that in the CTR (11.8 ± 3.7) (N-EOs vs. CTR; *p* < 0.05), but not in the S-EOs diet. These observations will provide a basis to advance current knowledge on the anatomy and digestive physiology of this species in relation to pro-heath feeds.

## 1. Introduction

In the last decade, the application of natural feed additives has been able to support optimal gut health and function, thus enhancing growth, feed utilization, and disease prevention in the whole aquaculture sector [1,2]. Essential oils (EOs) are extracted from plants raw materials and contain compounds produced during plant secondary metabolism. They are natural multicomponent systems of volatile, lipophilic, odoriferous, and liquid substances, obtained from complex mixtures of low-molecular-weight substances [3,4]. EOs contain compounds that are responsible for antimicrobial, antibacterial, anti-oxidant, antiviral, and antimitotic properties [5,6]. As a result, EOs have been the focus of aquaculture studies due to their diverse properties, and they are good candidates as they enhance the health, growth, and welfare of the fish [3].

In no-mammalian vertebrates, one cell type, the oxyntopeptic cells (OPs), secretes both hydrochloric acid and pepsinogen into the lumen to initiate protein digestion [7,8,9,10]. The hydrochloric acid promotes the conversion of pepsinogen into pepsin, an efficient proteolytic enzyme [9,11,12]. The gastric proton pump, H^+^/K^+^-ATPase, is responsible for stomach luminal acidification in vertebrates and is a characteristic feature of the gastric gland [13]. In addition to the H^+^K^+^-ATPase, Na^+^K^+^-ATPase in the gastric mucosa was also detected in vertebrates (including humans) [14,15]. The Na^+^K^+^-ATPase was found in correspondence of the parietal cells [14,16,17,18,19].

Ghrelin (GHR) is a 28-amino-acid peptide [20,21,22] that is involved in the control of energy homeostasis and increases food intake in mammals [23,24,25,26,27]. In fish, GHR mRNA is highly expressed in the stomach/gut, and moderate levels are detected in the brain [28,29,30]. GHR enteroendocrine cells (EECs) were detected in various tissues of no-mammalian vertebrates such as in the hypothalamus and stomach [30,31,32], as well as in the gastrointestinal tract of chicken [33], in the stomach of turtle [34,35], in the stomach of rainbow trout [36] and in the gut of zebrafish [37].

Unlike GHR, somatostatin (SOM) inhibits food intake, promotes catabolic processes (e.g., mobilization of stored lipid and carbohydrate) [38,39], and promotes the reduction of basal and/or stimulated gastric acid secretion [40]. SOM EECs have been documented in the alimentary canal of northern pike (*Esox lucius*) [41], milkfish (*Chanos chanos*) [42], and the predatory longnose gar (*Lepisosteus osseus*) [43].

In order to assess the gut health status of fish in relation to different feeding and nutritional strategies, the intestine has been largely considered the main target tissue for histological evaluation including determination of the morphological/morphometric characteristics of salmonids and other species of commercial interest [44,45,46,47,48,49]. Indeed, sparse attention was devoted to exploring the morphological features of the gastric mucosa in response to different feeding conditions. In addition, there is no detailed information on the histological features of the OPs in the European sea bass.

In this context, the aim of the present work was to investigate the presence and distribution of the OPs and GHR/SOM EECs in the gastric mucosa of the European sea bass fed diets supplemented with natural EOs and artificial EOs composed of active ingredients obtained by synthesis. Finally, we saw that Na^+^K^+^-ATPase was expressed in OPs in the same way as H^+^K^+^-ATPase (typical marker for mammalian parietal cells) and that, for this reason, it can be considered a valid marker for OPs.

## 2. Materials and Methods

### 2.1. Rearing Conditions and Tissue Sampling

A commercial diet (43% protein, 21% lipid, pellet diameter 4.0 mm VRM, Verona, Italy) was coated with HERBAL MIX^®^, a blend of essential oils, natural essential oils (N-EOs), or EOs obtained by synthesis (S-EOs). N-EOs contained a blend of natural essential oils of thyme, garlic, rosemary, and cinnamon, while S-EOs was a blend of thymol and carvacrol, diallyl sulfide, cineol, and cinnamaldehyde (main components of N-EOs). The concentration ratio between molecules was the same in N-EOs and in S-Eos, and the inclusion rate was 1000 g ton^−1^ for both blends. A diet without supplementation was kept as the control (CTR) group. Table 1 reports the fatty acid composition of the diets. The inclusion of the blends did not affect the overall fatty acid composition of the different diets.

European sea bass juveniles obtained from an Italian commercial hatchery were reared in a recirculating aquaculture system (RAS) at the Laboratory of Aquaculture, Department of Veterinary Medical Sciences of the University of Bologna, Cesenatico, Italy. At the beginning of the trial, 50 fish (75.0 ± 0.2 g) per tank were randomly distributed into 9, 900 L conical-bottom tanks provided with natural sea water (oxygen level 8.0 ± 1.0 mg L^−1^, temperature 23 ± 1.0 °C, salinity 25 g L^−1^). Each diet was fed twice a day to triplicate groups to satiation for 117 days using the overfeeding approach as described in Busti et al. [50]. Fish were individually weighed at the beginning and at the end of the trial.

At the end of the trial, four fish per tank (total of 36 specimens mean weight 270.0 ± 4.6 g) were sampled for gastrointestinal tract histology. After euthanasia with a lethal dose (300 mg L^−1^) of MS222, the stomach was gently isolated and fixed in 10% buffered formalin. After fixation, the stomach was cut symmetrically along the major axis to obtain two equal halves and embedded in paraffin.

### 2.2. Immunohistochemistry

Stomach sections were processed for single- and double-labeling immunofluorescence. Sections were deparaffinized, rehydrated, and incubated for 1 h in a humid chamber at room temperature (RT) with appropriate normal serum (5% normal goat or donkey serum) and 1% BSA diluted in PBS (phosphate buffered saline, 0.01 M pH 7.4) to reduce the nonspecific binding of the secondary antibodies. The sections were then incubated at 4 °C in a humid chamber for 24 h with the primary antibody rabbit anti-Na^+^K^+^-ATPase (Abcam, Cambridge, UK) diluted to 1:200. After washing, the sections were incubated at RT for 1 h with the goat anti-rabbit Alexa Fluor^®^ 488.

For the EECs, rat anti-somatostatin (Enzo Life Sciences, New York, NY, USA) diluted to 1:300, was used in association with mouse anti-ghrelin (Acris/OriGene, Herford, Germany) diluted to 1:300 in PBS. The sections were incubated at 4 °C overnight. After washing, the sections were incubated at RT for 1 h with donkey anti-rat Alexa Fluor^®^ 488 and donkey anti-mouse Alexa Fluor^®^ 594 secondary antibody (Invitrogen, CA, USA) diluted in PBS and then coverslipped with buffered glycerol, pH 8.6.

### 2.3. Threshold Binarization Method

In order to characterize the area occupied by the immunoreactive (-IR) OPs in the gastric mucosa, the following method was applied. The slides were scanned with the Nikon DS-Qi1Nc digital camera at 20× magnification, using NIS Elements software BR 4.20.01 (Nikon Instruments Europe BV, Amsterdam, The Netherlands). Automated Image Binarization was applied to an area of each selected gastric image by means of the NIS Elements software BR 4.20.01. Image binarization is a widely used method that allows to distinguish objects of interest from the background. Indeed, it determines a gray threshold and assigns each pixel of a digital image to one class (image objects). If it is a gray value, it is greater than the determined threshold compared to the other class (image background). Specifying correct threshold limits is a crucial procedure of the automated image analysis used to determine which pixels will or will not be included in the binary layer. In our case, using binarization, we were able to separate the pixels that represented the OP-IR cells of the gastric surface (brighter pixels) from those that represent the rest of the section (Figure 1A,B). The area of measurement can be restricted by a user-defined region of interest (ROI). Within the ROI, it is possible to binarize only the selected part and not include other parts of the section (Figure 1C). This allowed the quantification of the gastric surface covered by OP-IR cell area. The gastric morphometric assessments were performed in a blind fashion by two investigators.

### 2.4. Antibody Specificity

Specificity for GHR and SOM antibodies was demonstrated by the lack of immunostaining when the antibodies were pre-adsorbed with an excess of the homologous peptide. The Na^+^K^+^-ATPase antibody is specific for zebrafish. Furthermore, the recognized antigenic sequence of the Na^+^K^+^-ATPase antibody has a structural homology of 95% with that of sea bass. In addition, omission of the primary antibody excluded inappropriate binding of the secondary antibody.

### 2.5. Validation of the Na^+^K^+^-ATPase/H^+^K^+^-ATPase Antibodies as a Marker of Oxyntopeptic Cells (OPs)

Serial sections (4 µm thick) of sea bass stomach were used to validate the use of anti-Na^+^K^+^-ATPase antibody as a marker of OPs cells. In one section, the primary antibody H^+^K^+^-ATPase (Aviva System, San Diego, CA, USA) was used, while the other section was incubated with the Na^+^K^+^-ATPase primary antibody. Subsequently, the sections were incubated with donkey anti-rabbit Alexa Fluor^®^ 488. A total overlap of the OPs was observed (Figure 2). In addition, in the esophagus–stomach and stomach–intestine junctions, it was observed that the OPs tended to decrease until disappearance: this feature has been highlighted with both ATPase antibodies (Figure 2A–D).

### 2.6. Morphometric Evaluations

Preparations were examined on a Nikon Eclipse Ni microscope, and the images were recorded with a Nikon DS-Fi2 (for ordinary histology) and Nikon DS-Qi1Nc (for immunofluorescence) digital camera and NIS Elements software BR 4.20.01 (Nikon Instruments Europe BV, Amsterdam, The Netherlands). Slight adjustments to contrast and brightness were made using Corel Photo Paint, whereas the figure panels were prepared using Corel Draw (Corel Photo Paint and Corel Draw, Ottawa, ON, Canada). The 20× objective was used for the morphometric evaluation. In the gastric mucosa, the area occupied by the OPs-IR in 4.1 mm^2^ (0.410 × 10 fields) was measured by binarization (described above). Furthermore, in the gastric mucosa EECs, the number of GHR- and SOM-IRs in 4.1 mm^2^ were counted.

### 2.7. Statistical Analysis

All fish growth data are presented as mean ± standard deviation (SD). The tank was used as the experimental unit for analyzing growth performances. Data were analyzed by a one-way ANOVA and Tukey’s post hoc test. The differences among treatments were considered significant at *p* < 0.05.

The values obtained from the OP-IR area and the EEC number were grouped for each experimental group (CTR, N-EOs and S-EOs), and the means were calculated. Results were expressed as mean ± SD. Data were analyzed by one-way ANOVA (GraphPad Prism 4, GraphPad Software, Inc., La Jolla, CA, USA): we considered the experimental group as the main effect. In addition, means were subsequently separated by using Tukey—HSD test. A *p* value < 0.05 was considered statistically significant.

The gastric morphometric assessments were performed in a blind fashion by two investigators.

## 3. Results

Data on growth performances (final body weight and specific growth rate, SGR), feed intake (FI), and feed conversion rate (FCR) at the end of the trial are summarized in Table 2. No significant differences were observed in final body weight, SGR, FCR, and FI during the overall trial (*p* < 0.05).

The gastric mucosa is lined by a simple columnar epithelium composed of poorly stained epithelial cells with a central elongated nucleus: these elongated cells (mucins cells) were positioned above the gastric glands. Below the gastric glands, the presence of the lamina muscularis mucosae limits the mucosa from the submucosa. The submucosa is composed of loose connective tissue without the presence of glands. The muscular layer presents circular internal and longitudinal external bundle orientations.

By immunofluorescence, we observed OPs in all parts of the stomach: these cells showed intense immunoreactivity. The OPs were distributed along the adenomere of the simple tubular gastric glands (Figure 3). The immunoreactivity of the OPs was interrupted in the transition from esophageal epithelium to the gastric mucosa (Figure 4A–D). Similarly, in correspondence with the passage from gastric mucosa to the first part of the intestine, the positivity of the gastric glands was interrupted.

The highest density of the oxyntopeptic IR area was in the CTR group (0.66 mm^2^ ± 0.1). Oxyntopeptic IR area was significantly reduced in the N-EO diet (0.22 mm^2^ ± 1, CTR vs. N-EOs, *p* < 0.005), while in the S-EO diet (0.39 mm^2^ ± 1) a decreasing trend was observed (Figure 5).

EECs were mainly distributed over the glandular adenomeres: few cells were located along the glandular adenomere. Generally, the EEC cells are intermingled between mucous cells and, in some cases, tend to reach the endoluminal side (Figure 6A–D). Some SOM- or GHR-IR cells had the morphological appearance of “open-type” EECs with an elongated (“pear-like”), homogenous cytoplasm and two cytoplasmic prolongations, one reaching the lumen and the other the basal lamina. Other SOM or GHR-IR cells had the “closed-type” EEC appearance with a round shape without cytoplasmic prolongations. In particular, the EEC-IR cells located in the endoluminal side showed an “open-type” appearance, while those located in inner part of the mucosa displayed a “closed-type” shape (Figure 6C,D). We observed that SOM-IR cells co-expressed GHR and vice versa.

In the gastric mucosa, the N-EOs group (15.6 ± 4.2) exhibited a significant change in the mean number of SOM-IR cells compared to the CTR group (CTR 11.8 ± 3.7) (CTR vs. N-EOs; *p* < 0.05), while there were no significant differences in the mean number of the SOM-IR cells with respect to the N-EO diet (13.8 ± 3.4). We did not observe significant differences regarding the mean number of GHR-IR cells in the three experimental groups (CTR 11.9 ± 6, N-EO diet 13.8 ± 6.4 and S-EO diet 14.2 ± 7.8, respectively) (Figure 7). The percentage of colocalized IR cells/total of GHR-IR cells was 34% in the CTR (438/1297), while the percentage decreased in the N-EOs (28%, 397/1430) and S-EOs (28.3%, 394/1391) groups. Similarly, the percentage of colocalized IR cells/total of SOM-IR cells was 34% in the CTR (438/1288), while the percentage decreased in the N-EOs (28.6%, 397/1387) and S-EOs (25.9%, 394/1519).

## 4. Discussion

Previous studies have demonstrated the efficacy of herbal extracts on fish to attenuate stress response [1], improved immune system, enhanced gut tissue integrity [51,52,53], and increase feed digestibility and fish growth performance [1,54]. In this context, EOs have been studied for their several beneficial characteristics, such as antibacterial and anti-oxidant properties [55], and their ability to improve the feed conversion ratio by improved feed palatability [56,57,58], feed digestibility, and nutrient transport [57]. In addition, excess use of various antibiotics, hormones, and other synthetic drugs to control diseases and improve fish growth in aquaculture is the reason behind the emergence of drug-resistant bacteria, suppressed immunity in the host, and production of toxic substances harmful to the environment and human health [59]. For this reason, of late, the World Health Organization (WHO) encourages supplemented diets incorporated with medicinal herbs or plants that minimize the use of chemicals in the diet of fish [52].

In the present study, supplementing a basic diet with EOs containing thymus, rosemary, cinnamon, and garlic reduced the expression of OPs and increased the number of SOM and GHR-IR EECs in the gastric mucosa. In the literature, no author reports were found on the morphometric quantification (IR area and/or number) of OPs and EECs in the fish gastric mucosa. For this reason, we have made several assumptions.

To explain the reduction in OPs, natural small molecules produced from medicinal plants have been used for a long time to treat and prevent various pathologies such as peptic ulcer. Several phytocomponents such as flavonoids, tannins, terpenoids, and saponin have been reported in distinct anti-ulcer findings as possible gastro-protective agents [59]. However, some of these phytocomponents (i.e., tannin and saponin) are also known as potential antinutritional factor in fish species [60].

Thymus was used in the prevention/treatment of some gastrointestinal disorders. Several researchers report that thyme contains numerous phenolic compounds, especially thymol and carvacrol, which are found in its essential oil. Additionally, in wild thyme, many other abundant phenolic compounds have been found such as caffeic and rosmarinic acid derivatives. The anti-ulcerogenic effect of thymus extracts was demonstrated in rats stimulated with ulcer-inducing substances (e.g., HCL/ethanol, indomethacin) [59,61,62].

Rosemary, which is used in folk medicine, has many therapeutic properties: antifungal, antiviral, antibacterial, anti-inflammatory, antitumor, antithrombotic, antinociceptive, antidepressant, anti-ulcerogenic, and anti-oxidant activities [63,64,65,66,67,68]. Two groups of compounds are primarily responsible for the biological activity of this plant, the volatile fraction and phenolic constituents as rosmarinic acid [66] and fractions of flavonoids and diterpenes, which are structural derivatives of carnosic acid [67]. Amaral et al. [69] report that rosemary extracts play a protective action in ethanol-induced gastric ulcers in rats. Similar results on the anti-ulcerogenic activity of crude hydroalcoholic extract of *Rosmarinus officinalis* were obtained by Dias et al. [70] in rats.

Cinnamon is a traditional herb used for many diseases, and it has effects as an anti-oxidant, anti-inflammatory, antispasmodic, and anti-ulcerative agent. In the rat, the oral administration of cinnamon aqueous extracts for two weeks significantly improved gastric juice volume and decreased the gastric juice acidity and the gastric ulcer index [71]. In this regard, in the rat, intragastric administration of *Oleum cinnamomi* reduced gastric pH levels: the authors indicated that *Oleum cinnamomi* prevents ulcerative lesions and has beneficial effects on the gastric mucosa [71].

The beneficial effects of garlic on decreasing blood pressure, triglyceride levels, and oxidative activities and its anticarcinogenic, antibacterial, antifungal, and antiviral properties have been proved [72,73]. Garlic has abundant chemical compounds such as allicin, alliin, S-allyl cysteines, thiacremonone, diallyl-disulfide, diallyl sulfide, and others. Lee et al. [74] indicate that diallyl disulfide, a secondary organosulfur compound derived from garlic, prevents gastric mucosal damage induced by acute ethanol administration in rats. In this regard, the gastroprotective effects of garlic extract were shown by El-Ashmawy and El-Bahrawy [75] in rats treated with indomethacin-induced gastric ulcers.

Some authors claim that mammalian parietal cells to be functionally more capable of secreting acid than OPs present in fish and lower vertebrates [10,75,76,77]. For this reason, it is conceivable that essential oil compounds performed a similar action to that of acidifiers by inhibiting acid secretion.

Dietary acidifiers (organic and inorganic acids) have been broadly applied worldwide in the diets of animals (in order to replace antibiotic growth promoters), because of their potential to reduce both gastrointestinal-pH [78] and parietal cells [79,80]. Some researchers also claimed that dietary acidifiers in the feed of fish reduce the pH in the stomach, which helps improve pepsin activity, enhancing the protein metabolism and mineral intake of the intestines [10,80,81,82]. Previous studies have shown that European sea bass maintain a slightly acidic gastric pH (4.5–5) during fasting followed by a strong acidification (pH below 3) stimulated by the ingestion of food [82], possibly indicating the specific need for this species to reach a low gastric pH for optimal pepsin activity. However, a recent study has also hypothesized that the reduction of feed buffering capacity induced by the low fish meal content of current aquafeed formulations might lead to relatively low pH in the intestinal tract with consequences on feed utilization [83].

SOM acts as a potent inhibitor of gastric acid secretion. In gastrin gene knockout mice, a reduction in the number of parietal cells was observed, whereas the EEC number was not affected by gene deletion [84,85,86,87]. In our study, we showed an increase in SOM-IR cells in fish fed a diet supplemented with EOs. It is plausible that EOs reduce acid secretion by the stimulation of EECs (probably D cells). In mammals, the stimulatory effects of gastrin, histamine, and acetylcholine tightly regulate gastric acid secretion and the inhibitory actions of SOM on their respective receptors located in the parietal cells [87]. When gastric pH becomes too low, SOM secretion increases to inhibit not only acid production by parietal cells but also gastrin secretion by G cells [88]. In the piglet, Mazzoni et al. [80] observed an increase of the SOM-IR cells after feed supplementation of sodium butyrate in the post-weaning period. In the rat, intraperitoneal administration with thymoquinone significantly increased the number of SOM-positive cells [89], while, in piglets, intragastric administration of thymol upregulated SOM and SOM receptor (SSTR1 and SSTR2) genes in the gastric mucosa [90].

Another hypothesis could be that the phytocompounds, contained in the EOs, could have interacted with the OPs and EECs by means of transient receptor potential (TRP) channels. TRPV1 and TRPV4 receptor families displayed a ubiquitous distribution in mammals [91], zebrafish, and sea bass [92,93]. In the rainbow trout, TRPV1 (and TRPV4) is distributed in several organs, but its expression in the intestine was twofold higher than in the other districts such as retina, brain, pineal organ, spleen, heart, and blood cells [94].

TRPV1 (so-called “capsaicin receptors”) plays a key role in many other sensory functions and in detecting a large array of noxious stimuli and was also found in vagal, splanchnic, and pelvic visceral afferents, implicated in gastrointestinal mechanosensory functions and visceral hypersensitivity [95]. In addition, TRPV1 is expressed in parietal cells [96], endocrine G cells [97], gastric epithelial cells, as well as the esophageal, small intestinal, and colonic epithelial cells [97,98,99,100]. Besides the endogenous agents, TRPV1 is activated by several spices, such as capsaicin, cinnamaldehyde, allyl-isothiocyanate, and allicin [99,100,101].

Another hypothesis may be that the different components of EOs may have modulated gastrointestinal bacteria communities. The normal microbiota of the gastrointestinal tract surfaces contains saprophytic and potential pathogenic bacteria species, and both types are capable of multiplying and infecting the fish when conditions become favorable. Under normal conditions, fish maintain a dynamic microbial equilibrium in defense against these potential invaders using a repertoire of innate and specific defense mechanisms [102,103,104]. This bacterial community can modulate expression of genes in the digestive tract involved in the stimulation of epithelial proliferation, promotion of nutrient metabolism, and innate immune responses, while preventing the potential development of intestinal disorders and imbalances in intestinal homoeostasis [104]. Due to the chemical diversity and possible interactions among the molecules, EOs not only may modulate gut bacterial composition by their effects directly on the bacterial cell, but they also can affect the host in a number of other ways, mainly modulating the immune and other physiological responses. Recently, in European sea bass, a dietary blend of organic acid (citric acid, sorbic acid) and essential oils (thymol and vanillin) was able to induce a potential functional reconfiguration of the gut microbiome, promoting a significant decrease in several inflammation-promoting and homeostatic functions [50].

Regarding GHR, several authors report that phytochemical compounds of essential oils (e.g., cinnamaldehyde) decrease ghrelin secretion in mouse ghrelinoma 3-1 cell lines but, at the same time, upregulated the ghrelin gene expression [105]. Conversely, we observed an increase (not significant) of the GHR-IR cells in the gastric mucosa: the increase of the GHR is a positive aspect considering the ample evidence that GHR is an orexigenic peptide in several fish species. We do not have an explanation for the results obtained; probably the mechanisms that regulate the expression, presence, and distribution of GHR in the gastrointestinal tracts of fish are different from those in mammals.

In mammals, acid secretion in the stomach is mediated by parietal cells. The gastric H^+^K^+^-ATPase, a member of the P2-type ATPase family, is the integral membrane protein responsible for gastric acid secretion. P-type ATPases comprise five groups: Type I ATPases, Type II ATPases (Ca^2+^-ATPases, Na^+^K^+^-ATPases and H^+^K^+^-ATPases), Type III ATPases, Type IV ATPases, and Type V ATPases [106,107,108]. In addition to the H^+^-K^+^-ATPase, Na^+^K^+^-ATPase was also detected in the gastric mucosa in vertebrates including humans. Some authors have found Na^+^K^+^-ATPase in the gastric mucosa, in correspondence to parietal cells [14,16,17,18] and oxyntopeptic cells [19,109]. Present in all animal cells, Na^+^K^+^-ATPases appear at a higher concentration and more actively in seawater Teleosts [110].

In the present study, we have shown colocalization between Na^+^K^+^-ATPase and H^+^K^+^-ATPase in the sea bass OPs. On the other hand, Gonçalves et al. [109] showed, by means double immunofluorescence, Na^+^K^+^-ATPase/H^+^K^+^-ATPase co-expression in the gastrointestinal tract of some Cypriniformes.

## 5. Conclusions

For the first time, it has been shown in sea bass that the administration of a diet supplemented with the EO blend HERBAL MIX^®^ affects the distribution of OP, SOM, and GHR-IR cells in the gastric mucosa. It is possible that the EOs carry out directly or indirectly (by means the SOM EECs) an acidifying-like action.

In intensive and semi-intensive farming, it has long been known that various biotic and abiotic factors, as well as aquaculture procedures (handling, transport, or stocking density), activated a stress system that induces negative effects on different physiological processes in fish (growth, reproduction, and immunity).

Before reaching the intestine, the food undergoes numerous transformations within the stomach. In this context, EOs could represent a promising strategic alternative method to antibiotics/chemicals for maintaining and promoting health, as well as preventing and potentially treating some diseases and/or improving growth also in the gastric context.

Further studies are needed to gain more insights to understand the mechanisms of action of various EOs on the morphology of the fish gastric mucosa. The observations obtained in this study will provide a basis for a better understanding of the digestive physiology and help pathologists and nutritionists in future studies on diet and diseases affecting this species.

## Figures and Tables

**Figure 1 animals-11-03401-f001:**
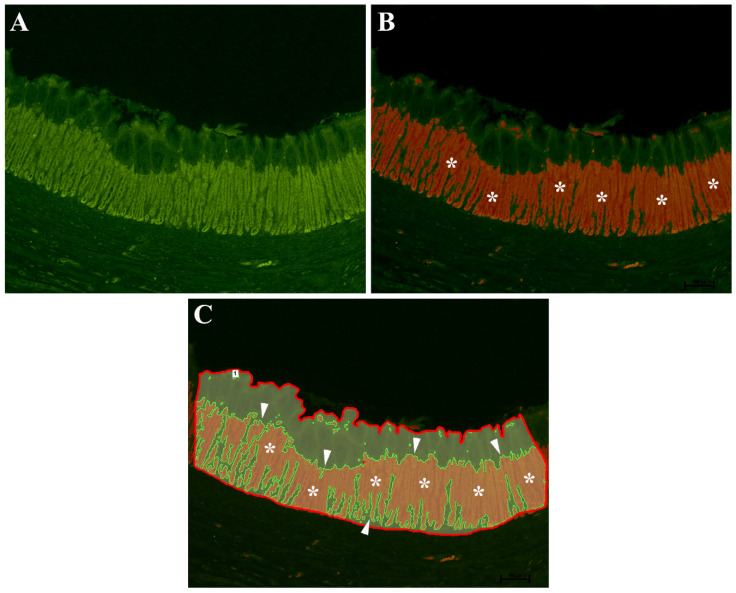
Threshold binarization method of European sea bass gastric mucosa. (**A**) Original acquired image. (**B**) Specific threshold to highlight the OP-IR cell area (asterisks). (**C**) Represents the region of interest (ROI) defined by the red line within which the binarized area is colored in red (asterisks) and delimited by a green line (arrowheads).

**Figure 2 animals-11-03401-f002:**
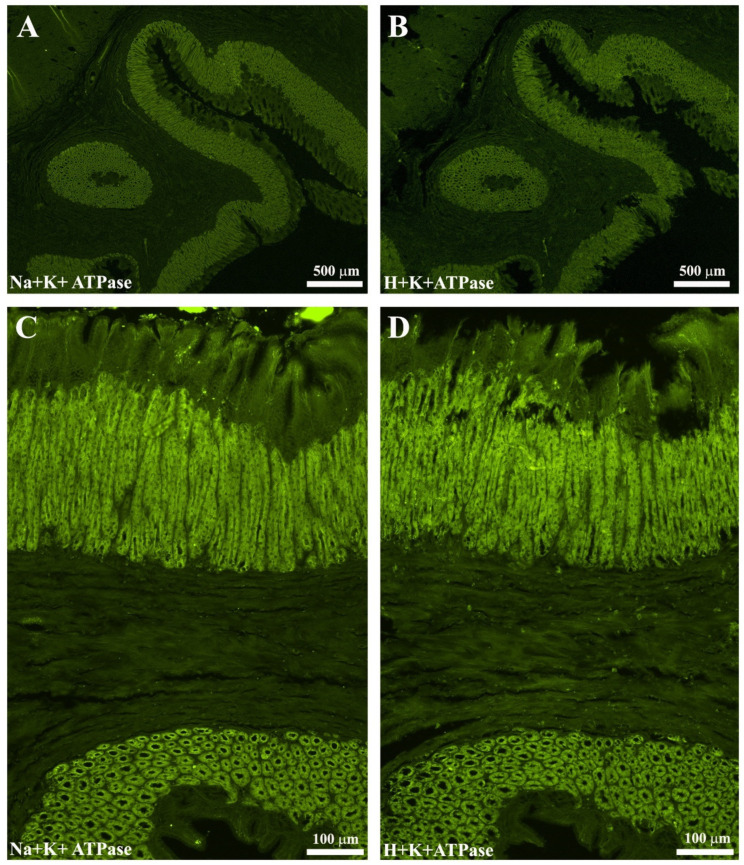
The images (**A**–**D**) show, in serial sections, a total overlapping of the oxyntopeptic-IR cells in the gastric mucosa both using Na^+^K^+^-ATPase (**A**,**C**) and H^+^K^+^-ATPase (**B**,**D**) primary antibodies.

**Figure 3 animals-11-03401-f003:**
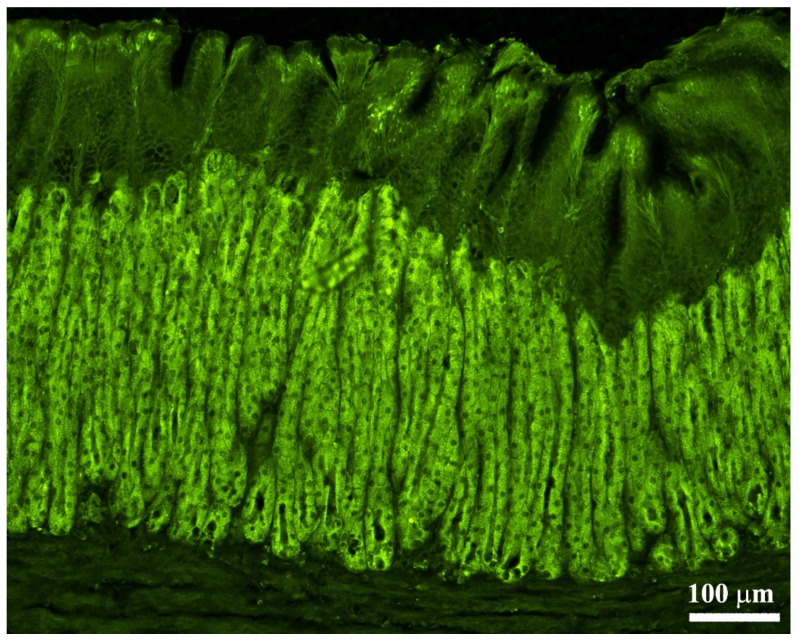
Localization of oxyntopeptic (OP) immunoreactive cells marked with Na^+^K^+^-ATPase antibody in the European sea bass gastric mucosa. The positive OP cells were observed along the tubular-like structure.

**Figure 4 animals-11-03401-f004:**
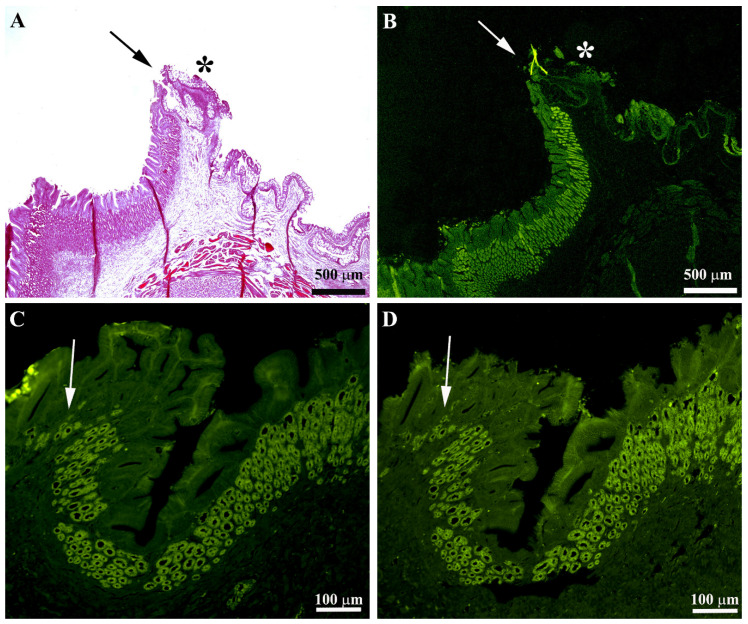
Gastric mucosa of the European sea bass. The images show the transition from esophagus to stomach in serial sections stained with hematoxylin–eosin (**A**) and Na^+^K^+^-ATPase (**B**). The simple esophageal columnar epithelium (asterisks) abruptly passes to the gastric mucosa with typical gastric glands (arrows). Images (**C**,**D**) show that the presence and distribution of the oxyntopeptic cells at the esophagus–stomach junction tend to decrease until they disappear (arrows): this feature was highlighted in both serial sections stained with Na^+^K^+^-ATPase and H^+^K^+^-ATPase antibody.

**Figure 5 animals-11-03401-f005:**
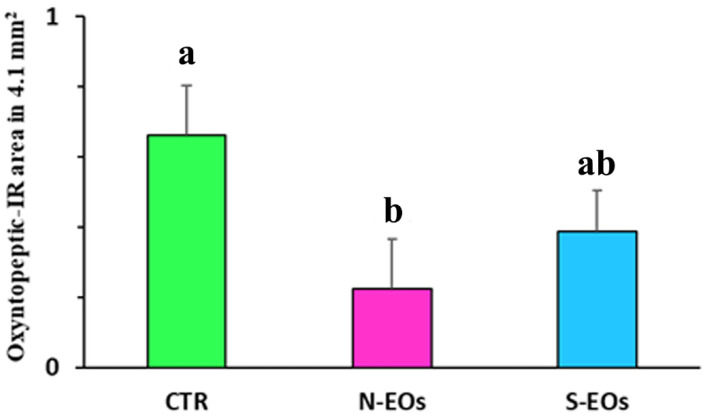
Graph showing the oxyntopeptic cells’ immunoreactive area in the sea bass gastric mucosa. CTR (control group), N-EOs (natural EOs), and S-EOs (EOs obtained by synthesis). Different letters (a and b) indicate significantly different mean values at *p* < 0.01. Values are expressed as mean + SD.

**Figure 6 animals-11-03401-f006:**
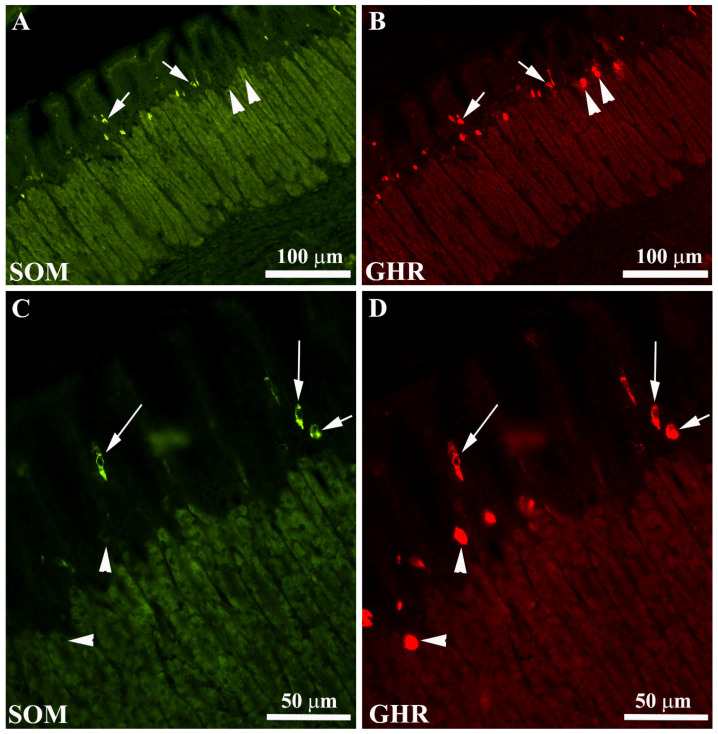
Localization of somatostatin (SOM) (**A**,**C**) and ghrelin (GHR) (**B**,**D**) enteroendocrine cells (EECs) in the European sea bass gastric mucosa. Some EECs co-expressing SOM/GHR-IRs (**A**,**B**, arrows). The arrowheads in (**A**,**B**) indicate GHR-IR cells (**B**) not containing SOM-IR (**A**). In some cases, both SOM and GHR-IR cells show a typical “open-type” EEC morphology (**C**,**D**, long arrows), while other SOM and GHR-IR cells were found lying close to the basal lamina of the glands and exhibiting typical “closed-type” EEC morphology (**C**,**D**, short arrows). Even in these higher-magnification images, some GHR-IR EECs were negative for SOM (**C**,**D**, arrowheads).

**Figure 7 animals-11-03401-f007:**
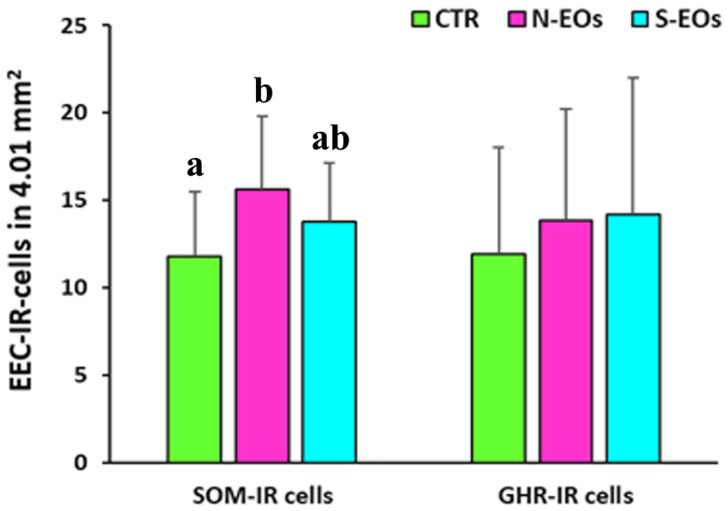
Graph showing the number of the somatostatin (SOM) and ghrelin (GHR) enteroendocrine (EEC) immunoreactive (IR) cells in the sea bass gastric mucosa. CTR (control group), N-EOs (natural EOs), and S-EOs (EOs obtained by synthesis). Different letters (a and b) indicate significantly different mean values at *p* < 0.05. Value are expressed as mean + SD.

**Table 1 animals-11-03401-t001:** Fatty acid composition of the three experimental diets.

Fatty Acid Composition (g/100 g)	CTR	N-EOs	S-EOs
Caprinic acid (10:0)		0.006 ± 0.001	0.006 ± 0.001
Lauric acid (12:0)	0.027 ± 0.006	0.031 ± 0.007	0.028 ± 0.006
Myristic acid (14:0)	0.163 ± 0.035	0.146 ± 0.031	0.150 ± 0.032
Pentadecanoic acid (15:0)	0.020 ± 0.004	0.019 ± 0.004	0.019 ± 0.004
Palmitic acid (16:0)	1.570 ± 0.190	1.550 ± 0.180	1.540 ± 0.180
Isoheptadecanoic acid (17:0 iso)	0.013 ± 0.003		0.011 ± 0.002
Hexadecenoic acid (16:1)	0.220 ± 0.630	0.193 ± 0.041	0.210 ± 0.630
14-Methylhexadecanoic acid (17:0 anteiso)	0.016 ± 0.003	0.015 ± 0.003	0.014 ± 0.003
Margaric acid (17:0)	0.026 ± 0.005	0.025 ± 0.005	0.024 ± 0.005
Heptadecenoic acid (17:1)	0.028 ± 0.004	0.029 ± 0.005	0.029 ± 0.004
Stearic acid (18:0)	0.425 ± 0.060	0.422 ± 0.061	0.418 ± 0.060
Octadecenoic acid (18:1)	8.250 ± 0.680	9.090 ± 0.760	8.930 ± 0.750
Octadecadienoic acid (18:2)	3.900 ± 0.370	4.070 ± 0.380	3.960 ± 0.380
Arachidic acid (20:0)	0.090 ± 0.019	0.091 ± 0.020	0.090 ± 0.019
Octadecatrienoic acid (18:3)	1.550 ± 0.180	1.440 ± 0.170	1.480 ± 0.170
Eicosenoic acid (20:1)	0.446 ± 0.063	0.374 ± 0.056	0.395 ± 0.058
Stearidonic acid (18:4 n-3)	0.050 ± 0.011	0.043 ± 0.009	0.044 ± 0.009
Behenic acid (22:0)	0.047 ± 0.010	0.050 ± 0.011	0.048 ± 0.010
Docosanoic acid (22:1)	0.267 ± 0.042	0.229 ± 0.036	0.234 ± 0.038
Lignoceric acid (24:0)	0.062 ± 0.013	0.061 ± 0.013	0.037 ± 0.008
Polyunsaturated fatty acids (>C20)	0.386 ± 0.049	0.342 ± 0.046	0.337 ± 0.046
Polyunsaturated fatty acids	6.210 ± 0.420	6.160 ± 0.420	6.100 ± 0.420
Monounsaturated fatty acids	9.260 ± 0.800	9.960 ± 0.760	9.840 ± 0.850
Saturated fatty acids	2.470 ± 0.210	2.420 ± 0.200	2.400 ± 0.200
Fatty acids ratios			
Polyunsaturated fatty acids/monounsaturated fatty acids	0.671 ± 0.074	0.618 ± 0.064	0.620 ± 0.069
Polyunsaturated fats/saturated fatty acids	2.510 ± 0.280	2.550 ± 0.280	2.540 ± 0.280
Volatile organic acids (mg/kg)			
Acetic acid	627 ± 94	700 ± 110	700 ± 110
Butyric acid	67 ± 22	62 ± 22	66 ± 22

**Table 2 animals-11-03401-t002:** Growth performance of European sea bass fed experimental diets over 117 days.

Experimental Diet	CTR	N-EOs	S-EOs	*p* Value
IBW (g)	75.3 ± 2.88	74.9 ± 1.54	74.9 ± 2.42	0.835
FBW (g)	274.2 ± 7.81	267.9 ± 3.49	263.6 ± 4.19	0.137
SGR	1.10 ± 0.05	1.09 ± 0.03	1.07 ± 0.02	0.529
FI	1.47 ± 0.05	1.41 ± 0.02	1.42 ± 0.03	0.158
FCR	1.52 ± 0.08	1.54 ± 0.06	1.52 ± 0.02	0.894

Data are given as the mean (*n* = 3) ± SD. No significant differences among treatments (One-way ANOVA, *p* > 0.05). IBW = initial body weight. FBW = final body weight. SGR = specific growth rate (% day^−1^) = 100 × (ln FBW − ln IBW)/days. FI = feed intake (% average body weight^−1^, AWB day^−1^) = ((100 × total ingestion)/(ABW))/days. FCR = feed conversion rate = feed intake/weight gain.

## Data Availability

The data presented in this study are available on request from the corresponding author.

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
