# Peer review of "Effect of Essential Oils on the Oxyntopeptic Cells and Somatostatin and Ghrelin Immunoreactive Cells in the European Sea Bass (Dicentrarchus labrax) Gastric Mucosa"

_animals, 2021, doi:10.3390/ani11123401_

Round 1

Reviewer 1 Report

The manuscript firstly estimates the effect of essential oils on the oxyntopeptic, somatostatin and ghrelin immunoreactive cells of the stomach in European sea bass. The experiment was well conducted and obtained data sets were well presented. I have just a few minor comments.

L171: remove “and”

L275-319:

Some components included in the essential oils have been known as an anti-nutritional factor for fish (Francis et al., Aquaculture 199, 197-227, 2001). Not only the positive effects seen in mammals, but also negative effects of the components in fish should be mentioned in the discussion.

Author Response

We thank the reviewer for his/her comment. We have removed "end" from line 171. 

We thank the reviewer for his/her comment and following the suggestion we we added, in the discussion section, a sentence about some components included in essential oils that have been known as anti-nutritional factors for fish (Francis et al., Aquaculture 199, 197-227, 2001). New lines 284-285. 

Reviewer 2 Report

Thanks to the authors for their answers and corrections. Just one question: the error remains in the scientific name

Author Response

We thank the reviewer for reporting the Latin scientific  name of the sea bass in the title. I had already reported this anomaly to the Editor. I will do the report again this time and, if necessary, when the proof arrives. 

This manuscript is a resubmission of an earlier submission. The following is a list of the peer review reports and author responses from that submission.

Round 1

Reviewer 1 Report

AQUACULTURE-D-21-00836

The data contained in this manuscript are potentially interesting and might contribute aquaculture research aria, but the presentation must be improved considerably. Especially, the methodology is very unclear, so I could not judge if the study was fair or not. Thus, this manuscript is not acceptable for publication in its present form.

Shortly, some specific points are mentioned in below.

Diets:

Detail of the diets (oil) must be mentioned.

For example,

・added amount of the oils to the diet

・mixture ratio of the oils for N-EOs

・detail of the “synthetic oil”

・Fatty acid composition of the three diets

・Fatty acid composition of the blended essential oils and the synthetic oil.

Growth:

Growth performance of fish fed the experimental diets must be shown.

Statistical analysis:

Multiple comparison test should be performed.

Author Response

The data contained in this manuscript are potentially interesting and might contribute aquaculture research aria, but the presentation must be improved considerably. Especially, the methodology is very unclear, so I could not judge if the study was fair or not. Thus, this manuscript is not acceptable for publication in its present form.

Shortly, some specific points are mentioned in below.

Diets:

Detail of the diets (oil) must be mentioned.

For example,

・added amount of the oils to the diet

We thank the reviewer for his/her comment: inclusion rate was 1000g/ton

・mixture ratio of the oils for N-Eos

We thank the reviewer for his/her comment. The concentration ratio between molecules was the same in N-EOs and in S-EOs. Tecnessenze S.r.l. cannot divulgate the exact mixture ratio because it is a proprietary information. The company apologize for that but it is sure that reviewer can understand its need to protect its data.

・detail of the “synthetic oil”

We thank the reviewer for his/her comment.  N-EOs contained a blend of natural essential oils of thyme, garlic, rosemary and cinnamon, while S-EOs a blend of thymol and carvacrol, diallyl sulphide, cineol and cinnamaldehyde (main components of N-EOs).  

・Fatty acid composition of the three diets

The fatty acid composition of the different diets was added to the manuscript in Table 1. Blends inclusion did not affect fatty acid composition or nutritional values of diets.

・Fatty acid composition of the blended essential oils and the synthetic oil.

We thank the reviewer for his/her comment. Blends inclusion did not affect fatty acid composition or nutritional values of diets.

Growth:

Growth performance of fish fed the experimental diets must be shown.

We thank the reviewer for his/her comment, growth performance has been added in Table 2 and in the results section.

Reviewer 2 Report

The paper approaches a very interesting subject about the influence of diet ingredients on activity of specific cells of gastric epithelium. The manuscript is well planned and written. I think it is worthy of being published but needs some issues to improve.

The title is clear and provides appropriate information on the content of the manuscript, but I don't understand the reason for writing the scientific name of European seabass as Dicen-Trarchus Labrax. It is commonly accepted Dicentrarchus labrax (in italics).

We are not talking about a fish nutrition experiment, but the only information described in relation to the diet used to feed the fish is the protein and lipid content (line 88). What was the proportion of natural essential oils included in the diets? What was the proportion and chemical profile of synthetic essential oils?

The standard deviations both in the initial weight (line 95) and especially in the weight at the end of the trial (lines 100 and 101) are incredibly low. In any case, were the fish caught randomly?

I would add a new section on the methodology for the contents of lines 176 to 182: 2.7. Statistical analysis.

Were there differences between tanks in the diets? (line 176) First question to answer before grouping the results of each diet.

If the morphometric evaluation was performed by two people, I think that a general linear model with diet as a fixed factor and the researcher as a random factor is more adequate.

I think that the information and references from lines 270 to 299 should be included in the introduction to justify the aims of the research. The presentation of the results obtained by other authors requires discussion and comparison with our results. If not, it is a simple review.

The conclusion is very poor: “the EOs affected some gastric mucosa morphological characteristic”. What is the extent of this qualitative and quantitative affection? What are the possible health impacts on the fish?

Author Response

Reviewer 2

The paper approaches a very interesting subject about the influence of diet ingredients on activity of specific cells of gastric epithelium. The manuscript is well planned and written. I think it is worthy of being published but needs some issues to improve.

The title is clear and provides appropriate information on the content of the manuscript, but I don't understand the reason for writing the scientific name of European seabass as Dicen-Trarchus Labrax. It is commonly accepted Dicentrarchus labrax (in italics).

We thank the reviewer for his/her comment. I don't know what happened, but in the version of the manuscript we submitted we wrote the Latin name Dicentrarchus labrax and not Dicen-Trarchus Labrax. We double checked for safety.

We are not talking about a fish nutrition experiment, but the only information described in relation to the diet used to feed the fish is the protein and lipid content (line 88). What was the proportion of natural essential oils included in the diets? What was the proportion and chemical profile of synthetic essential oils?

We thank the reviewer for his/her comment. The concentration ratio between molecules was the same in N-EOs and in S-EOs. The inclusion rate was 1000 g/ton for both blends.  N-EOs contained a blend of natural essential oils of thyme, garlic, rosemary and cinnamon, while S-EOs a blend of thymol and carvacrol, diallyl sulphide, cineol and cinnamaldehyde (main components of N-EOs).  

The standard deviations both in the initial weight (line 95) and especially in the weight at the end of the trial (lines 100 and 101) are incredibly low. In any case, were the fish caught randomly?

We thank the reviewer for his/her comment. Yes. Fish were caught randomly at the beginning and also at the end of the trial.

I would add a new section on the methodology for the contents of lines 176 to 182: 2.7. Statistical analysis.

We thank the reviewer for his/her comment and following the suggestion we have added to new section called “2.7. Statistical analysis” (new line 193).

Were there differences between tanks in the diets? (line 176) First question to answer before grouping the results of each diet.

No significant differences were observed among dietary treatments.

We thank the reviewer for his/her comment. We caught 4 fish per tank for a total of 12 fish for each experimental diet. Statistical analysis did not show any difference between the data obtained in each tank. 

If the morphometric evaluation was performed by two people, I think that a general linear model with diet as a fixed factor and the researcher as a random factor is more adequate.

We thank the reviewer for his/her comment and following the suggestion we applied the one way ANOVA and considered the experimental group as the main effect (new line 196-203). We have changed the statistical differences in the results (new lines 222 and 253). In addition, we performed an orthogonal contrast test (data not show) which confirmed what was previously highlighted with the one way ANOVA test. Finally, regarding the morphometric evaluations carried out by two researchers, the two people in charge of the morphometric evaluations were previously trained and numerous cross-checks were carried out on the data obtained.

I think that the information and references from lines 270 to 299 should be included in the introduction to justify the aims of the research. The presentation of the results obtained by other authors requires discussion and comparison with our results. If not, it is a simple review.

We thank the reviewer for his/her comment. For the first time, a quantification (by means of morphometry) of the immunoreactive oxyntopeptic, somatostatin and ghrelin cells in the sea bass gastric mucosa was performed. For this reason, we have not found anything in the bibliography regarding the evaluation of the presence and distribution of oxyntopeptic, somatostatin and ghrelin immunoreactive cells in the gastric mucosa of sea bass (and of any other fish species) and, consequently, we have not found anything related to the effect of essential oils on the gastric mucosa. On the basis of these considerations we have reported various hypotheses to try to explain the morphometric results obtained. In particular, we hypothesized that essential oils could have carried out directly and indirectly (through somatostatin enteroendocrine cells) an acidifying-like action. It is for this reason that we have included in the discussions the description of some characteristics of the 4 botanical compounds used (thyme, rosemary, cinnamon and garlic) focusing on their respective anti-ulcerogenic properties. 

The conclusion is very poor: “the EOs affected some gastric mucosa morphological characteristic”. What is the extent of this qualitative and quantitative affection? What are the possible health impacts on the fish?

We thank the reviewer for his/her comment and following the suggestion we have modified and implemented the conclusions.

 Statistical analysis:

Multiple comparison test should be performed.

We thank the reviewer for his/her comment. We applied Tukey-HSD test and we inserted the letters that indicate the statistical differences in the histograms (Figure 5 and 7, respectively).
